# Glutamic acid is a carrier for hydrazine during the biosyntheses of fosfazinomycin and kinamycin

Kwo-Kwang A. Wang[1,2], Tai L. Ng[3], Peng Wang[3,5], Zedu Huang[1,2,6],
Emily P. Balskus [3] & Wilfred A. van der Donk [1,2,4]

Fosfazinomycin and kinamycin are natural products that contain nitrogen–nitrogen (N–N) bonds but that are otherwise structurally unrelated. Despite their considerable structural differences, their biosynthetic gene clusters share a set of genes predicted to facilitate N–N bond formation. In this study, we show that for both compounds, one of the nitrogen atoms in the N–N bond originates from nitrous acid. Furthermore, we show that for both compounds, an acetylhydrazine biosynthetic synthon is generated first and then funneled via a glutamyl carrier into the respective biosynthetic pathways. Therefore, unlike other pathways to N–N bond-containing natural products wherein the N–N bond is formed directly on a biosynthetic intermediate, during the biosyntheses of fosfazinomycin, kinamycin, and related compounds, the N–N bond is made in an independent pathway that forms a branch of a convergent route to structurally complex natural products.

---

[1] Department of Chemistry, University of Illinois at Urbana-Champaign, Urbana 61801 IL, USA. [2] Carl R. Woese Institute for Genomic Biology, University of Illinois at Urbana-Champaign, Urbana 61801 IL, USA. [3] Department of Chemistry and Chemical Biology, Harvard University, Cambridge 02138 MA, USA. [4] Howard Hughes Medical Institute, Chevy Chase 20815 MD, USA. [5] Present address: Red & Charline McCombs Institute for the Early Detection and Treatment of Cancer, University of Texas MD Anderson Cancer Center, Houston 77030 TX, USA. [6] Present address: Department of Chemistry, Fudan University, Shanghai 200438-6789, China. These authors contributed equally: Tai L. Ng and Peng Wang. Correspondence and requests for materials should be addressed to E.P.B. (email: balskus@chemistry.harvard.edu) or to W.A.v.d D. (email: vddonk@illinois.edu)

More than 200 natural products containing nitrogen–nitrogen (N–N) bonds have been identified with various bioactivities (Fig. 1a)[1]. For instance, streptozotocin, a nitrosamine-containing compound, exhibits cytotoxic activity and is currently deployed clinically in the treatment of gastrointestinal and pancreatic cancers[2], and valanimycin, featuring an azoxy group, is a broad-spectrum antimicrobial that also displays cytotoxicity towards mouse leukemia cells[1]. Other examples are azamerone containing a pyridazine, kutzneride with a piperazic acid[2,3], the fosfazinomycins having a phosphonohydrazide[4], and a group of molecules including cremeomycin, the lomaiviticins, and the kinamycins containing diazo groups (Fig. 1a)[1].

Despite the diversity of N–N bonds found in natural products, relatively little is currently known about their biosyntheses. Based on the available biosynthetic gene clusters, the biosynthetic logic of N–N bond formation appears to be at least partially shared among certain groups of compounds. Recently, N–N bond formation was reconstituted in vitro for kutzneride, showing that N5 on ornithine is oxidized to give N5-hydroxyornithine by a flavin-dependent enzyme before the N–N bond is formed by a heme protein (Supplementary Fig. 1a)[5,6]. Similarly, in the biosynthesis of s56-p1, lysine is oxided to N6-hydroxylysine, and the N6-hydroxyl is then joined with the carboxyl group of glycine. The N–N bond is subsequently formed through intramolecular nucleophilic attack by the glycine amino group (Supplementary Fig. 1b)[7]. Thus, for both of these natural products, one nitrogen is activated to a more electrophilic species and then reacted intramolecularly with a nucleophilic nitrogen.

A different strategy towards N–N bond formation features the formal reaction of a nitrogen-containing intermediate with nitrous acid. For instance, during cremeomycin biosynthesis, CreE, a flavin-dependent monooxygenase, oxidizes aspartic acid to nitrosuccinic acid (1, Fig. 1b), and a lyase, CreD, subsequently liberates nitrous acid[8]. Nitrous acid then partakes in diazotization with a primary aromatic amine on an advanced intermediate to form the N–N bond in a reaction catalyzed by CreM, a fatty acid-coenzyme A ligase homolog (Supplementary Fig. 1c)[9,10]. Recently, we reported that the fosfazinomycin biosynthetic enzymes FzmM (UniProtKB A0A0N0UQ79) and FzmL (UniProtKB U5YN81), homologs of CreE and CreD, respectively, can also produce nitrous acid from aspartic acid with N-hydroxyaspartic acid (2, Fig. 1b) as an intermediate[11]. Moreover, we showed that at least one of the nitrogen atoms in the phosphonohydrazide moiety of fosfazinomycin is derived from aspartic acid. Interestingly, a set of five genes is conserved between the biosynthetic gene clusters of fosfazinomycin (fzm) and kinamycin (kin) (Fig. 1c), suggesting that the two compounds may share biosynthetic steps. These genes are predicted to encode homologs of a glutamine synthetase (FzmN/KinL), an amidase (FzmO/KinK), a hypothetical protein (FzmP/KinJ), an N-acetyltransferase (FzmQ/KinN), and an adenylosuccinate lyase (FzmR/KinM) (Supplementary Table 1)[12,13]. The roles of these five enzymes in installing the N–N bond into two structurally divergent compounds are not known. In our prior work concerning the biosynthesis of fosfazinomycin, we reconstituted in vitro the activities of FzmQ (UniProtKB U5YN85) and FzmR (UniProtKB A0A0M9CPV0). FzmQ performs the acetylation of hydrazinosuccinic acid (3) to yield N-acetylhydrazinosuccinic acid (4, Fig. 1b), and FzmR catalyzes an elimination reaction of 4 to afford fumaric acid and acetylhydrazine[11].

In this study we show via isotope labeling experiments that one of the nitrogen atoms in the N–N bonds of both fosfazinomycin and kinamycin originates from nitrous acid. Additional feeding experiments and reconstitution of enzymatic activities confirm the intermediacies of 2, 4, and acetylhydrazine in both

biosynthetic pathways. Moreover, we demonstrate that after the N–N bond is formed, the hydrazine moiety is installed onto the side chain carboxyl group of glutamic acid. These results reveal that during the biosyntheses of both fosfazinomycin and kinamycin, the N–N bond is formed from aspartic acid as a discrete and separate synthon before being shuttled into the two respective biosynthetic pathways using glutamic acid as carrier. Thus, these pathways are fundamentally different from the biosynthetic routes to kutzneride and cremeomycin in which the N–N bonds are formed directly on the scaffolds of the final natural product.

## Results

**Feeding fosfazinomycin biosynthetic intermediates.** In previous work, we showed using stable isotope labeling experiments that at least one of the nitrogen atoms in the phosphonohydrazide linkage in fosfazinomycin is derived from aspartic acid[11]. Furthermore, reconstitution of the activities of FzmM and FzmL demonstrated that they produce nitrous acid from aspartic acid; however, we also found that under certain conditions, FzmM converted aspartic acid to N-hydroxyaspartic acid (2)[11]. In light of the recent work on kutzneride and cremeomycin, it was unclear whether the generation of 2 by FzmM is a direct intermediate in N–N bond formation (similar to kutzneride or s56-p1, Supplementary Fig. 1ab) or if 2 is only an intermediate in the oxidation of aspartic acid to 1 (similar to cremeomycin, Supplementary Fig. 1c)[5,9]. In order to investigate these possibilities, we performed isotope labeling studies in the native producing organism of fosfazinomycin, Streptomyces sp. NRRL S-149. First, the organism was cultivated in a minimal medium with $^{15}NH_4Cl$ and $^{15}N$-aspartic acid as the only two sources of nitrogen, and analysis by liquid chromatography-high resolution mass spectrometry (LC-HRMS) of the spent medium revealed a singly charged peak ($[M + H]^+ = 461.1957$) corresponding to fosfazinomycin A containing seven $^{15}N$ atoms (Fig. 2a). Cultures grown in uniformly $^{15}N$-labeled medium were then fed sodium nitrite at natural isotopic abundance (i.e., unlabeled). LC-HRMS analysis of the spent medium revealed a clear increase in the peak corresponding to $[M + H]^+ = 460.1996$ (Fig. 2b), one mass unit lower than uniformly $^{15}N$-labeled fosfazinomycin. Integration of the peak areas showed that ~30% of the product contains a single $^{14}N$ atom originating from nitrite. We next fed unlabeled 2 to Streptomyces sp. NRRL S-149 grown in medium containing $^{15}NH_4Cl$ and $^{15}N$-aspartic acid as the only two other sources of nitrogen. As with nitrite, a significant peak with $[M + H]^+ = 460.1993$ was observed (Fig. 2c), corresponding to a single incorporation (20%) of $^{14}N$ into fosfazinomycin.

We next performed tandem mass spectrometry (MS/MS) analysis to determine if the $^{14}N$ atoms from nitrite and 2 were incorporated into the hydrazide linkage of fosfazinomycin. First, as a control, uniformly $^{15}N$-labeled fosfazinomycin ($[M + H]^+ = 461.20$) was subjected to higher energy collisional dissociation (HCD), and the product and fragment ions were assigned (Supplementary Fig. 2ab). Then, the spent medium from the culture cultivated with unlabeled sodium nitrite was analyzed focusing on a precursor ion ($[M + H]^+ = 460.20$) corresponding to singly $^{14}N$-labeled fosfazinomycin (Supplementary Fig. 2c). Analysis of the fragment ions indicated $^{14}N$-incorporation only if the fragment contained the N–N bond (ions g, j, and k, accented by blue stars). Likewise, in the sample fed with 2, the singly $^{14}N$-labeled fosfazinomycin (precursor ion $[M + H]^+ = 460.20$) was fragmented (Supplementary Fig. 2d). Again, only fragment ions containing the N–N bond (ions g, j, and k) incorporated $^{14}N$. Thus, these feeding experiments demonstrate that one of the nitrogens in the phosphonohydrazide linkage in fosfazinomycin is derived from nitrite. These data are also consistent with our

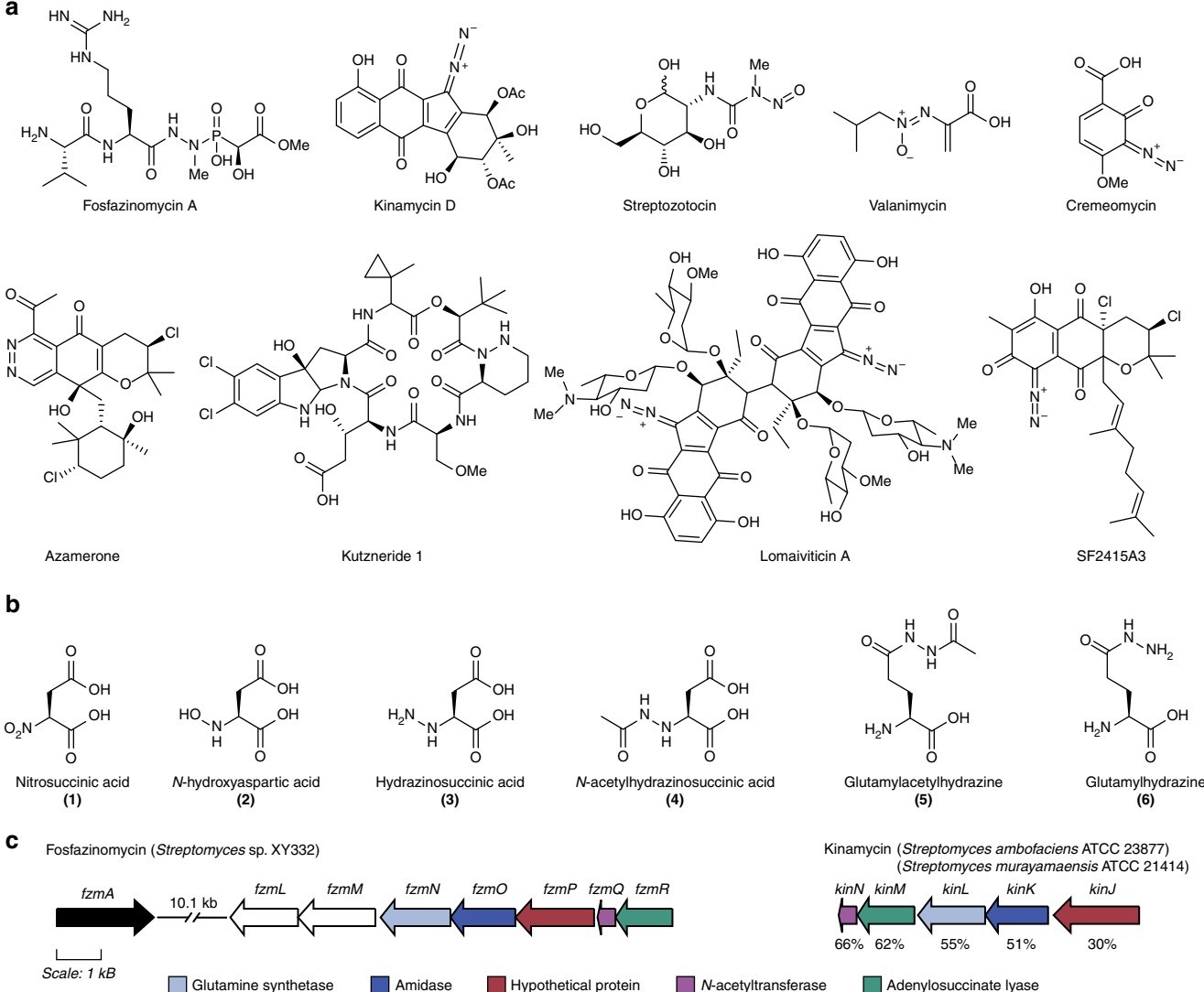

**Fig. 1** Structures and biosynthetic gene clusters of fosfazinomycin and kinamycin. **a** Structures of a select group of N–N bond-containing natural products. **b** Structures of the biosynthetic intermediates discussed in this study. **c** Selected segments of the biosynthetic gene clusters of fosfazinomycin and kinamycin that encompass the genes discussed in this study. The five conserved genes in the two clusters are colored based on annotations of their gene products using BLAST analysis: glutamine synthetase (blue-gray), amidase (dark blue), hypothetical protein (red), *N*-acetyltransferase (purple), and adenylosuccinate lyase (green). The amino acid identities between the homologous enzymes from the *fzm* cluster from *Streptomyces* sp. XY332 and the *kin* cluster from *Streptomyces murayamaensis* ATCC 21414 (FzmN/KinL, FzmO/KinK, FzmP/KinJ, FzmQ/KinN, and FzmR/KinM) are indicated below the *kin* cluster. *fzmL* and *fzmM*, which encode enzymes responsible for forming nitrous acid from aspartic acid, are colored in white

previous study that showed that nitrite is formed from Asp with **2** as an intermediate.

We next performed additional feeding experiments to confirm that **3** and acetylhydrazine are also intermediates in the fosfazinomycin biosynthetic pathway. When unlabeled **3** was fed to the producing organism grown in $^{15}$N-labeled medium, LC-HRMS analysis showed a large increase in a peak two mass units lower [M+H]$^+$ = 459.2013) than the uniformly $^{15}$N-labeled product (Fig. 2d). The data demonstrate about 50% double incorporation of $^{14}$N into fosfazinomycin. Similarly, when unlabeled acetylhydrazine was added to the $^{15}$N-labeled medium, a large increase in a peak two mass units lower ([M+H]$^+$ = 459.2015) relative to the control was observed, indicating two incorporations of $^{14}$N (40%) into fosfazinomycin A (Fig. 2e). MS/MS was then used to determine the location of the $^{14}$N atoms. Samples obtained from feeding **3** were analyzed by MS/MS using doubly $^{14}$N-labeled fosfazinomycin as the precursor ion ([M+H]$^+$ = 459.20) (Supplementary Fig. 2e). After HCD, only the

fragment ions that retained the N–N bond (ions f, g, j, and k indicated by the blue stars) exhibited a −2 Da mass shift compared to the control indicating selective incorporation of $^{14}$N into the N–N bond of fosfazinomycin. Likewise, when doubly $^{14}$N-labeled fosfazinomycin (precursor [M+H]$^+$ = 459.20) obtained from feeding acetylhydrazine was fragmented, only product or fragment ions containing the N–N bond contained two $^{14}$N atoms (Supplementary Fig. 2f). Thus, these data strongly suggest that **3** and acetylhydrazine are intermediates in the pathway and that both of their nitrogen atoms are incorporated selectively into the phosphonohydrazide linkage of fosfazinomycin, consistent with the demonstration that in vitro FzmQR convert **3** to acetylhydrazine[11].

**Nitrite labels the proximal diazo nitrogen in kinamycin.** As described in the introduction, the kinamycin biosynthetic gene cluster contains genes that are homologous to *fzmQR*, suggesting

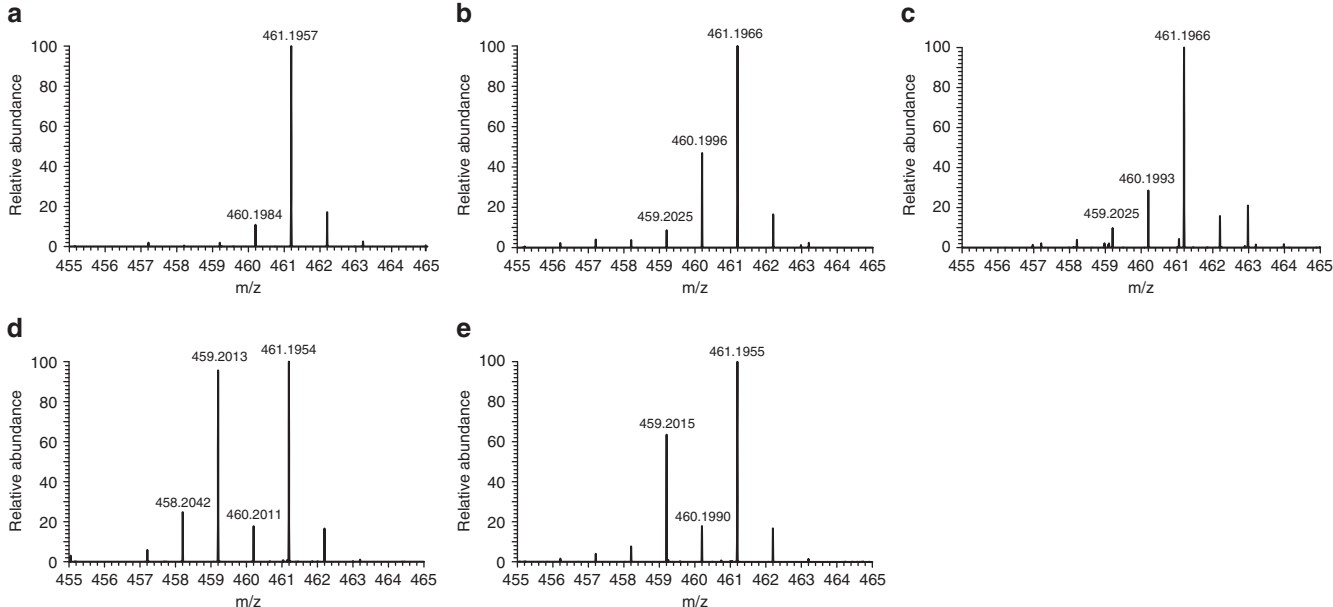

**Fig. 2** HRMS analysis of $^{14}$N-incorporation into $^{15}$N-labeled fosfazinomycin A. **a** Fosfazinomycin A produced by *Streptomyces* sp. NRRL S-149 cultivated in uniformly $^{15}$N-labeled medium. The calculated *m/z* ([M+H]$^+$) for uniformly $^{15}$N-labeled fosfazinomycin A is 461.1966. **b**, **c** Spectra from cells cultivated in uniformly $^{15}$N-labeled media and fed with either (**b**) NaNO$_2$ or (**c**) *N*-hydroxyaspartic acid (**2**). The calculated *m/z* ([M+H]$^+$) for fosfazinomycin A with a single $^{14}$N atom is 460.1996. **d**, **e** Spectra from cells grown in uniformly $^{15}$N-labeled media and fed with either **d** hydrazinosuccinic acid (**3**), or **e** acetylhydrazine. The calculated *m/z* ([M+H]$^+$) for $^{15}$N-fosfazinomycin A with two $^{14}$N atoms is 459.2025

that **3** and acetylhydrazine might also be intermediates in its biosynthesis. Furthermore, the biosynthesis of kinamycin D has been recently reconstituted in *S. albus* J1074, and its heterologous production has been demonstrated to be dependent on *alp2F* and *alp2G*, orthologs of *fzmM* and *fzmL*[14]. To investigate whether the products of FzmLM are incorporated into kinamycin, we performed a series of stable isotope feeding studies.

We first supplemented the fermentation medium of *S. murayamaensis* ATCC 21414, a prolific producer of kinamycin D[15], with 1 mM of $^{15}$N-nitrite (Fig. 3a). LC-HRMS analysis of the organic extracts from the $^{15}$N-nitrite-fed cultures revealed ~90% enrichment of singly labeled kinamycin D ([M+H]$^+$ = 456.1055) (Fig. 3b). After isolation of the natural product, $^{15}$N NMR spectroscopy showed that the proximal nitrogen of the diazo group was exclusively enriched (Fig. 3c). This outcome contrasts sharply with the results of previous $^{15}$N-nitrite feeding experiments performed with the producers of the diazo-containing secondary metabolite SF2415A3, in which only the distal diazo nitrogen atom was labeled by either $^{15}$N-nitrite or $^{15}$N-nitrate (Supplementary Fig 1d)[3]. We then tested whether $^{15}$N-nitrate could also label the diazo group of kinamycin D. After culturing *S. murayamaensis* in the presence of 1 mM $^{15}$N-nitrate, ~80% enrichment of the proximal nitrogen was observed (Fig. 3a–c). Diazo formation in the biosynthesis of kinamycin had previously been proposed to occur via a late-stage N–N bond formation involving an aminobenzo[*b*]fluorene biosynthetic precursor such as stealthin C, a biosynthetic logic that would parallel that of SF2415A3 (Supplementary Fig. 1de)[16]. The localization of $^{15}$N label to the proximal diazo nitrogen atom in our feeding experiments shows that this is not the case and supports the involvement of an alternative strategy for diazo installation in kinamycin biosynthesis.

**KinNM convert hydrazinosuccinic acid to acetylhydrazine.** To demonstrate that **3** and acetylhydrazine are also relevant intermediates in the biosynthesis of kinamycin, we first sequenced the

genome of *S. murayamaensis* ATCC 21414 and identified the homologs of FzmNOPQR, labeled as KinLKJNM, respectively. Then we expressed and purified KinN and KinM, which are homologs of FzmQ and FzmR, respectively (Fig. 1c). We observed the conversion of **3** to **4** by KinN in the presence of acetyl-CoA (Supplementary Fig. 3ab). Compound **4** was converted by KinM into acetylhydrazine and fumaric acid as monitored by $^1$H NMR spectroscopy and LC-HRMS (Supplementary Fig. 3ac). We next synthesized doubly-labeled $^{15}$N$_2$-acetylhydrazine and fed this substrate to *S. murayamaensis*. About 20% enrichment of both nitrogen atoms in kinamycin D was observed by LC-HRMS ([M+H]$^+$ = 457.1026) (Supplementary Fig. 3d). These results indicate a role for KinNM and acetylhydrazine in assembling the diazo group of kinamycin D.

**FzmN and KinL install acetylhydrazine onto glutamic acid.** Having confirmed that the nitrogen atoms involved in the hydrazide linkage of fosfazinomycin and the diazo group in kinamycin likely have common origins, we next focused on other enzymes encoded by the conserved five-gene cassette. FzmN (UniProtKB A0A0M9CP55) is a glutamine synthetase homolog, a class of enzymes that canonically catalyzes the formation of glutamine from glutamic acid and ammonia. We first tested whether FzmN could catalyze this prototypical reaction. FzmN was expressed recombinantly in *E. coli* with an *N*-terminal His$_6$-tag. The purified recombinant protein was then incubated with glutamic acid and ammonium chloride in the presence of ATP and MgCl$_2$ (Fig. 4a). When the reaction was analyzed by $^1$H NMR spectroscopy, a new peak was observed at 2.24 ppm that was absent when FzmN was omitted (Fig. 4b, c). This new resonance is consistent with the γ-protons on glutamine, and this assignment was confirmed by spiking with a glutamine standard.

FzmQ and FzmR convert **3** to acetylhydrazine[11], and the labeling studies described above establish that both of the nitrogen atoms from **3** and acetylhydrazine are incorporated

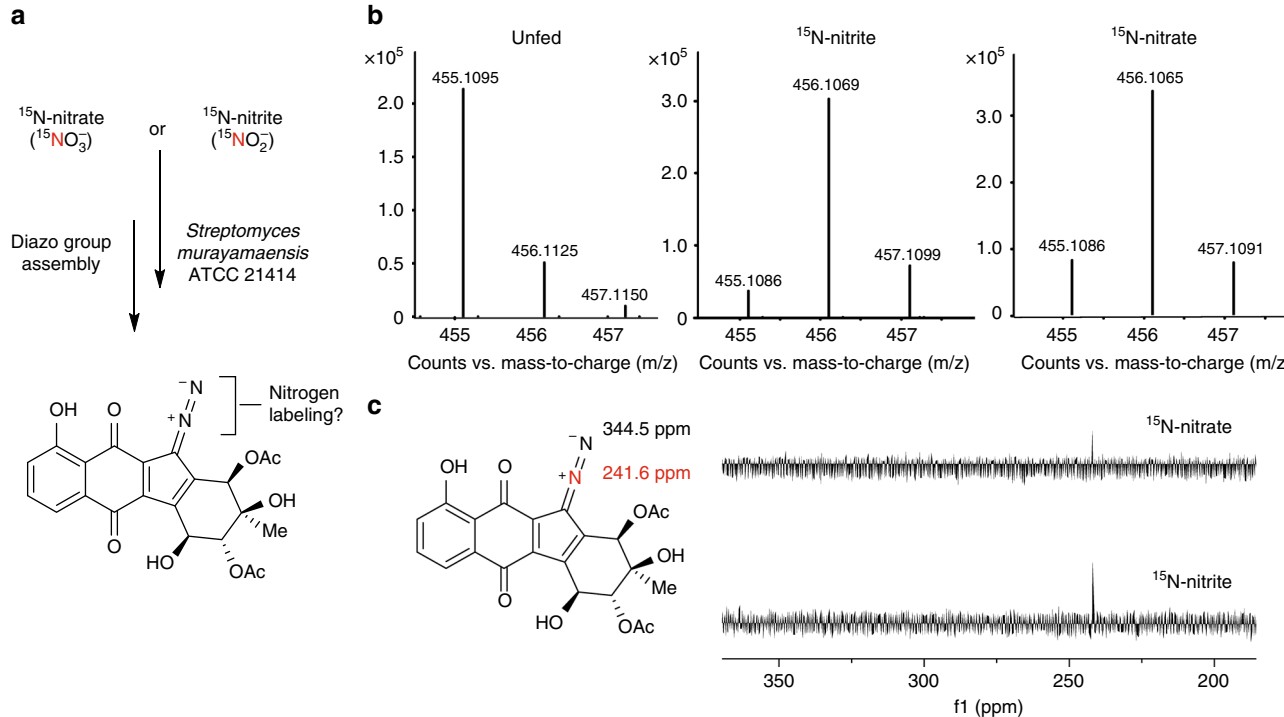

**Fig. 3** $^{15}$N-incorporation into kinamycin D by inorganic nitrogen sources. **a** Scheme for feeding experiments with *S. murayamaensis* using $^{15}$N-labeled nitrite and nitrate. **b** HRMS analysis of $^{15}$N incorporation into kinamycin D when no labeled nitrogen-containing salts, 1 mM $^{15}$N-nitrite, or 1 mM $^{15}$N-nitrate, were added to the fermentation culture of *S. murayamaensis*. The calculated *m/z* ([M+H]$^+$) for kinamycin D at natural abundance is 455.1091, $^{15}$N-kinamycin D is 456.1055, and $^{15}$N$_2$-kinamycin D is 457.1026. **c** $^{15}$N NMR analysis of kinamycin D isolated from fermentation cultures of *S. murayamaensis* revealed that $^{15}$N-nitrate and $^{15}$N-nitrite labeled the proximal nitrogen of the diazo group (highlighted in red)[30]

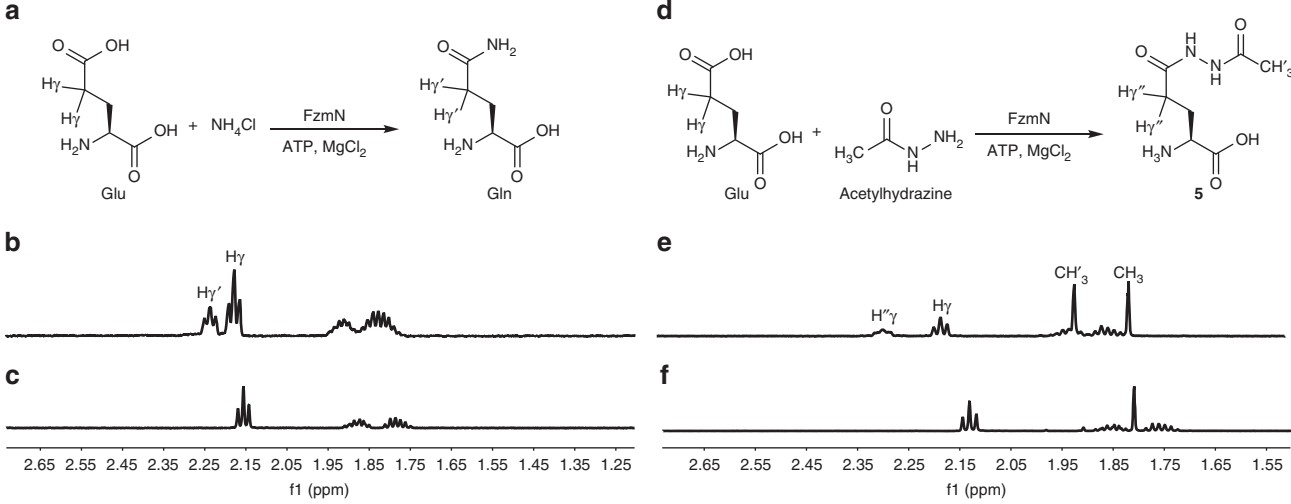

**Fig. 4** $^1$H NMR analysis of FzmN activity. **a** The canonical glutamine synthetase reaction catalyzed by FzmN. The γ-protons on the Glu substrate (Hγ) and Gln product (Hγ') are highlighted. **b** $^1$H NMR analysis of a reaction mixture containing FzmN, Glu, NH$_4$Cl, ATP, and MgCl$_2$. Peaks corresponding to the γ-protons on Glu and Gln are indicated. **c** $^1$H NMR analysis of the reaction mixture containing Glu, NH$_4$Cl, ATP, and MgCl$_2$ without FzmN. **d** The putative reaction catalyzed by FzmN during the biosynthesis of fosfazinomycin. The γ-protons on the Glu substrate (γ) and the glutamylacetylhydrazine (**5**) product (γ'') and the methyl protons on the acetylhydrazine substrate (CH$_3$) and **5** (CH'$_3$) are indicated. **e** $^1$H NMR analysis of the reaction mixture containing FzmN, Glu, acetylhydrazine, ATP, and MgCl$_2$. Resonances corresponding to the γ-protons and methyl protons on the substrates and the product are indicated. **f** $^1$H NMR analysis of the reaction mixture containing Glu, acetylhydrazine, ATP, and MgCl$_2$ without FzmN. The chemical shifts of several of the protons are sensitive to conditions (pH, salt), and hence spiking with authentic materials was used for assignments

selectively into the phosphonohydrazide linkage of fosfazinomycin. Thus, we reasoned that the next step in the biosynthetic pathway could involve acetylhydrazine as a substrate. Indeed, FzmN catalyzes the condensation of glutamic acid and acetylhydrazine to form glutamylacetylhydrazine (**5**) in the presence of ATP and MgCl$_2$ (Fig. 4d). When the reaction was analyzed by $^1$H NMR spectroscopy, two new resonances belonging to **5** at 2.28 ppm and at 1.92 ppm were observed that

were absent when FzmN was withheld from the reaction mixture (Fig. 4e, f).

Kinetic parameters were determined for FzmN using a coupled enzyme assay with pyruvate kinase and lactate dehydrogenase, measuring NADH consumption by UV-vis spectroscopy. First, the canonical glutamine synthetase reaction with ammonium chloride as a substrate was analyzed, yielding a $k_{cat}$ of $0.91 \pm 0.02\,s^{-1}$ and a $K_M$ of $17 \pm 1\,mM$ (Supplementary Fig. 4a) (the error was calculated from the standard deviation of triplicate runs). The FzmN reaction with acetylhydrazine displayed a similar $k_{cat}$ ($\sim 1.4\,s^{-1}$), but acetylhydrazine was such a good substrate that the $K_m$ could not be determined and is at least three orders of magnitude lower than the $K_m$ of ammonia ($K_{m,acetylhydrazine} < 10\,\mu M$) (Supplementary Fig. 4b). Together, these results strongly suggest that the condensation of acetylhydrazine and glutamic acid is the role of FzmN in the biosynthesis of fosfazinomycin.

We also tested KinL, the homolog of FzmN encoded in the *kin* biosynthetic gene cluster, for this ligase activity. While we were not able to obtain this enzyme in soluble active form, we confirmed its activity in vivo by feeding acetylhydrazine to *E. coli* expressing KinL (Supplementary Fig. 5a). Compound **5** was observed in culture supernatants only when KinL was expressed and acetylhydrazine was present in the media (Supplementary Fig. 5b). These results demonstrate that KinL possesses analogous reactivity to FzmN.

**FzmO forms glutamylhydrazine**. We next turned our attention towards FzmO (UniProtKB U5YQN5), an amidase homolog shared between the biosynthetic pathways of fosfazinomycin and kinamycin. Despite considerable effort, we were unable to find conditions to express recombinant FzmO solubly in *E. coli*. Thus, we endeavored to refold the protein in vitro. FzmO, expressed with an N-terminal His$_6$-tag, was extracted from inclusion bodies with buffers containing guanidine hydrochloride. After purification with nickel affinity chromatography, the protein was diluted into 216 buffers in 96-well plates, and conditions that resulted in solubilized FzmO were identified by measuring the turbidity of the wells with a plate reader[17]. The refolded soluble FzmO was then assessed for enzymatic activity. Glutamylacetylhydrazine (**5**), the product of the FzmN-catalyzed reaction, contains an acetyl group that would likely have to be lost prior to funneling the hydrazine synthon into fosfazinomycin biosynthesis. Likewise, in kinamycin biosynthesis, removal of the acetyl group would be necessary before the formation of the final product. Thus, we investigated whether FzmO, being an amidase homolog, might catalyze such a reaction. Refolded FzmO was incubated with **5**, and the products in the reaction mixture were derivatized with fluorenylmethyloxycarbonyl chloride (Fmoc-Cl) to facilitate reversed-phase LC-MS analysis (Fig. 5a). When the reaction mixture was analyzed by LC-MS, a mass consistent with glutamylhydrazine (**6**) was detected that was absent when FzmO was omitted from the reaction (Fig. 5b, c). To confirm the presence of **6**, a synthetic standard was prepared, and LC-MS analysis revealed that the synthetic material eluted with the same retention time as the product of the FzmO-catalyzed reaction (Fig. 5d). The assignment was then confirmed by spiking the synthetic standard into the reaction mixture (Fig. 5e). The FzmO-catalyzed hydrolysis appears to be selective for the acetylhydrazide linkage over the glutamylhydrazide bond since glutamic acid could not be detected after the reaction (Fig. 5f, g).

Further isotope labeling studies were then performed to assess the intermediacy of **6** in fosfazinomycin biosynthesis. Synthetic unlabeled **6** was provided to the producing organism, *Streptomyces* sp. NRRL S-149, cultured in uniformly $^{15}$N-labeled medium. HRMS analysis of the resulting spent medium revealed

a peak two mass units lower than uniformly $^{15}$N-labeled fosfazinomycin, indicating that two $^{14}$N atoms were indeed incorporated into fosfazinomycin at 40% enrichment (Fig. 5h). Tandem MS analysis on this doubly $^{14}$N-labeled fosfazinomycin (observed $[M+H]^+ = 459.2012$) suggests that the $^{14}$N atoms were incorporated selectively into the phosphonohydrazide moiety (Supplementary Fig. 2g).

The *kin* gene cluster encodes a homolog of FzmO, KinK. We demonstrated that **6**, the presumed product of KinK-mediated hydrolysis, is also an on-pathway intermediate in the biosynthesis of kinamycin D using feeding experiments. We synthesized $^{15}$N$_2$-L-glutamylhydrazine ($^{15}$N$_2$-**6**), wherein both of the nitrogen atoms in the hydrazine moiety are $^{15}$N-labeled, and supplemented the fermentation broth of *S. murayamensis* with this compound at a final concentration of $200\,\mu M$, as higher concentrations were found to have an adverse effect on growth and production of kinamycin D. We observed 60% enrichment of the diazo group in kinamycin D (Fig. 5i). This observation confirms that this functional group derives from a hydrazine-bound glutamyl scaffold.

**Reconstitution of FzmA activity**. Having reconstituted the production of **6** by FzmO, we next explored ways in which the hydrazine moiety of **6** could be transferred onto the carboxyl group of arginine since our previous work suggested that argininylhydrazine is an intermediate in the biosynthesis of fosfazinomycin[18]. Such a reaction would minimally require both the breakage and formation of a hydrazide linkage. FzmA (UniProtKB U5YQM4), an asparagine synthetase homolog, stood out as a prime candidate to catalyze this reaction. Bifunctional asparagine synthetases contain a glutaminase domain that hydrolyzes the side chain carboxamide of glutamine to liberate ammonia, which travels through a tunnel to a distal synthetase active site where aspartic acid is adenylated and reacted with the ammonia[19]. To assess enzymatic activity, FzmA was expressed recombinantly in *E. coli* bearing a C-terminal His$_6$-tag since the N-terminal cysteine residue is critical for catalysis in canonical asparagine synthetases. After the recombinant protein was incubated with **6**, the products of the reaction mixture were derivatized with Fmoc-Cl. Fmoc-Glu was detected by LC-MS (Supplementary Fig. 6a), whereas this product was not observed when FzmA was omitted from the reaction (Supplementary Fig. 6b). At present we have not been able to observe condensation of the hydrazine with arginine to form argininylhydrazine nor have we observed adenylation of Arg, but FzmA clearly has the ability to liberate hydrazine from glutamylhydrazine.

**Free hydrazine is not involved in kinamycin biosynthesis**. The fosfazinomycin and kinamycin biosynthetic pathways employ the key intermediate **6** in distinct ways. To construct the diazofluorene scaffold of the kinamycins, the hydrazine moiety of **6** must be hydrolyzed and transferred to a polyketide intermediate. We have previously reported that an electrophilic species generated from the biosynthetic intermediate dehydrorabelomycin by the flavin monooxygenase AlpJ/KinO1 can be attacked nonenzymatically by a variety of amine nucleophiles (Fig. 6)[20]. We envisioned that hydrazine liberated from **6** could attack this scaffold to generate a hydrazine adduct, which could be further oxidized to afford the diazofluorene. However, this reaction is unlikely to involve non-enzymatic addition of free hydrazine as we would have observed labeling of both the distal and proximal diazo nitrogen atoms in our $^{15}$N-nitrite and $^{15}$N-nitrate feeding experiments (Fig. 3).

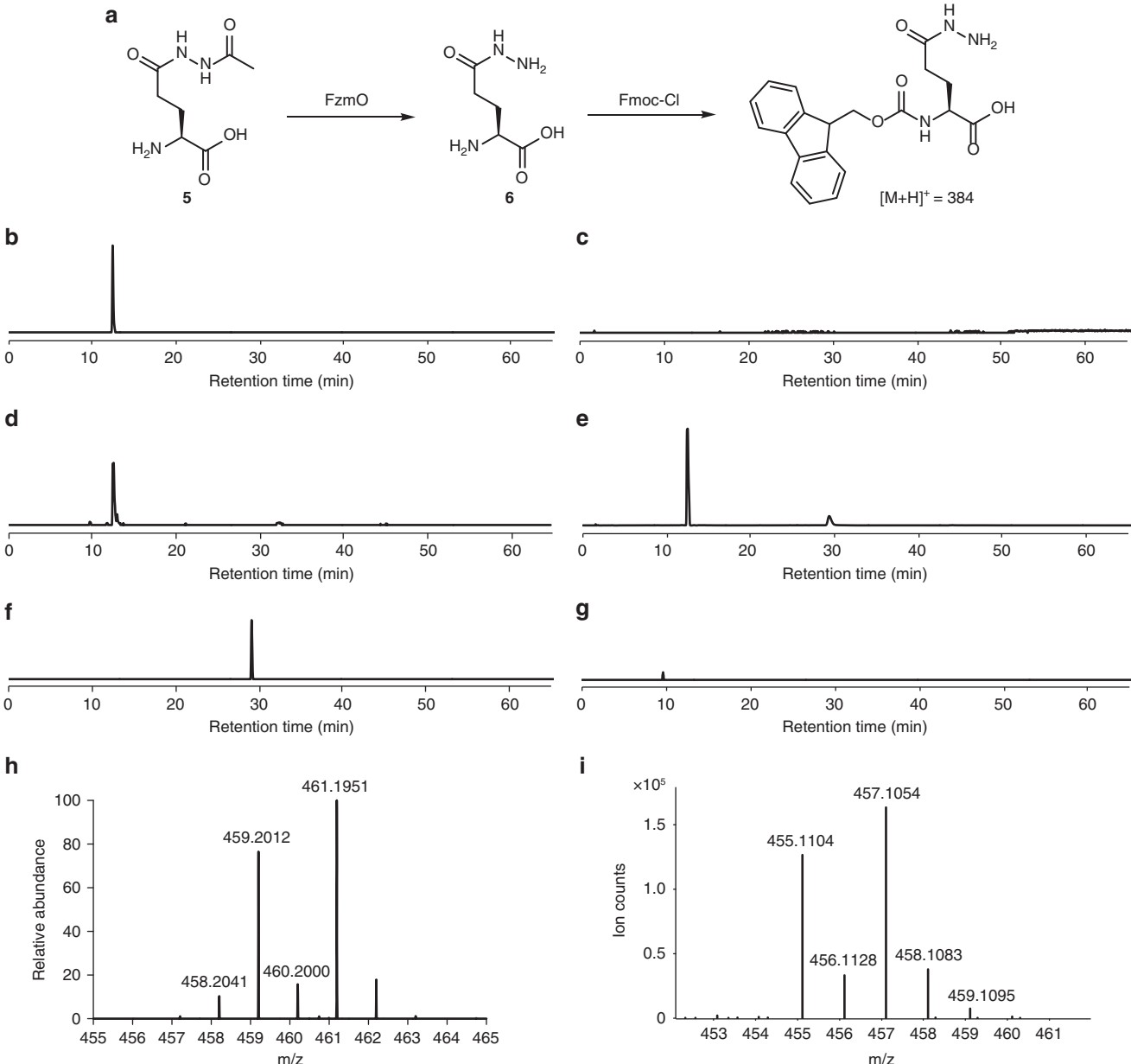

**Fig. 5** FzmO converts glutamylacetylhydrazine (**5**) to glutamylhydrazine (**6**). All samples were derivatized with Fmoc-Cl, and the mass of Fmoc-derivatized **6** ([M+H]$^+$ = 384) was monitored by LC-MS. **a** Deacetylation of **5** to **6** by FzmO in vitro. **b** Extracted ion chromatogram (EIC) for $m/z$ 384 of the reaction containing FzmO and **5**. **c** EIC for $m/z$ = 384 of a sample containing **5** without FzmO. **d** EIC for $m/z$ 384 of a synthetic standard of Fmoc-**6**. **e** EIC for $m/z$ 384 of a synthetic standard of Fmoc-**6** spiked into the reaction mixture containing FzmO and **5** in panel **b**. **f** EIC for $m/z$ 368 of Fmoc-derivatized Glu standard. **g** EIC for $m/z$ 368 of the reaction mixture of FzmO and glutamylacetylhydrazine. **h** HRMS analysis of $^{14}$N incorporation into $^{15}$N-labeled fosfazinomycin A in *Streptomyces* sp. NRRL S-149 fed with unlabeled **6**. The calculated $m/z$ ([M+H]$^+$) for $^{15}$N-fosfazinomycin A containing two $^{14}$N atoms is 459.2025. **i** HRMS analysis of $^{15}$N incorporation into kinamycin D from feeding 0.2 mM of $^{15}$N$_2$-**6** to cultures of *S. murayamaensis*. $^{15}$N$_2$-kinamycin D ([M+H]$^+$ = 457.1026)

To test whether free $^{15}$N$_2$-hydrazine can directly label kinamycin D, we added $^{15}$N$_2$-hydrazine to the fermentation broth of *S. murayamaensis* at a concentration of 0.2 mM, the same concentration at which we fed $^{15}$N$_2$-**6**. At this concentration, we observed 0% labeling (Supplementary Fig. 7a). At 1 mM concentration, we observed ~40% labeling of both diazo nitrogen atoms by mass spectrometry (Supplementary Fig. 7b). We hypothesized that KinL might exhibit promiscuous ligase activity, generating $^{15}$N$_2$-**6** from $^{15}$N$_2$-hydrazine directly in the producing strain. To test this proposal, we fed 1 mM hydrazine to *E. coli*

expressing KinL and observed production of **6** only when KinL was expressed and hydrazine was present (Supplementary Fig. 5c). Therefore, the labeling we observed from feeding $^{15}$N$_2$-hydrazine likely arises from its incorporation into **6** by KinL.

## Discussion

The in vivo feeding experiments presented here demonstrate that one of the nitrogen atoms in the N–N bonds of both fosfazinomycin and kinamycin is derived from nitrous acid, similar to

**Fig. 6** Proposed biosynthetic pathways for fosfazinomycin and kinamycin. Solid arrows indicate steps that have been reconstituted. Dashed lines indicate putative steps. Compound (**6**) (highlighted by the red box) acts as the carrier of the hydrazine synthon and is the branch point where the biosyntheses of fosfazinomycin and kinamycin diverge

findings in the biosynthesis of cremeomycin. But unlike in the biosynthesis of cremeomycin, in which the diazo group is formed late in the pathway from a pre-installed aromatic amine, we show that in the biosyntheses of fosfazinomycin and kinamycin, the N–N bond is made early and that the hydrazine functionality is carried through multiple enzymatic steps. Our feeding experiments corroborate the previously reconstituted in vitro activities of FzmQ and FzmR, thus firmly establishing that their physiological roles are indeed the acetylation of hydrazinosuccinic acid (**3**) and the elimination of acetylhydrazine from N-acetylhydrazinosuccinic acid (**4**), respectively (Fig. 6). Furthermore, the reconstitution of the activities of KinN and KinM, the homologs of FzmQ and FzmR in the kinamycin biosynthetic cluster, and the results of the feeding studies for kinamycin production illustrate that these steps are conserved in the biosynthetic pathways towards fosfazinomycin and kinamycin. These findings are surprising since previous studies had suggested that the N–N bond in the diazo group of kinamycin was fashioned from an aromatic nitrogen-containing precursor[16], analogous to the proposal for cremeomycin, but such a pathway is inconsistent with the labeling studies presented here.

To follow the fate of acetylhydrazine, we also reconstituted the activities of FzmN and FzmO, revealing that acetylhydrazine is first condensed onto the side chain carboxyl group of glutamic acid to form glutamylacetylhydrazine (**5**) before deacetylation to yield glutamylhydrazine (**6**). Our labeling studies in the native producing organisms of both fosfazinomycin and kinamycin corroborate that **6** is an intermediate in their biosyntheses. Glutamic acid is present in relatively high concentrations in the

cytoplasm of most bacteria including streptomycetes[21], and perhaps for that reason, glutamic acid has previously been demonstrated to be a common carrier molecule to effect many different types of transformations[22]. Most examples come from catabolism, but some have been uncovered in secondary metabolism. For instance, in the biosynthesis of butirosin, an intermediate conjugated to an acyl carrier protein (ACP), γ-aminobutyryl-ACP, is condensed onto the side chain of glutamic acid, and this glutamylated intermediate undergoes two further enzymatic transformations before deglutamylation[22–24].

In the fosfazinomycin biosynthetic pathway, we previously reported that argininylhydrazine is the substrate for the N-methyltransferase FzmH[18]. Thus, the next step in the biosynthesis after the formation of **6** likely involves the transfer of hydrazine from the side chain of glutamic acid onto the carboxyl group of arginine. FzmA, an asparagine synthetase homolog, might catalyze such a reaction (Fig. 6), and indeed, we have reconstituted the hydrolysis of **6** by FzmA, liberating hydrazine. While at present we have yet to reconstitute the enzymatic formation of argininylhydrazine by FzmA, there is precedent for asparagine synthetase homologs catalyzing the transfer of N-X moieties from glutamylated intermediates. A recent report demonstrated that the asparagine synthetase homolog TsnB9 catalyzes the transfer of hydroxylamine from the side chain of glutamic acid to the carboxyl group of an advanced intermediate during the course of trichostatin biosynthesis (Supplementary Fig. 1f)[25].

The mechanism of elaboration of **6** to arrive at the diazo group of kinamycin is likely more complicated. We envision two general scenarios through which hydrazine could be transferred from this

precursor to a polyketide scaffold. In the first scenario, **6** could form an adduct with a polyketide intermediate, and then a peptidase or an amidase-like enzyme could remove the amino acid via hydrolysis. Alternatively, an enzyme could hydrolyze **6**, generating hydrazine in its active site. Transfer of hydrazine to the polyketide scaffold would then be catalyzed by this enzyme or a partner transferase. Consistent with this latter proposal, both *kin* and lomaiviticin gene clusters possess a conserved amidotransferase-like enzyme, KinW. These potential routes, along with potential polyketide substrates, are summarized in Supplementary Fig. 8. Future studies will seek to identify the mechanism and enzymes involved in this process.

Regardless of the details of transfer of the hydrazine synthon to the polyketide scaffold, this work provides further support for a major revision in the proposed biosynthetic pathway for kinamycin and other diazofluorene polyketides. A previous study had suggested that stealthin C is a biosynthetic intermediate to kinamycin on the basis of isotope labeling studies; however, that study measured the incorporation of deuterated stealthin C, and only low percentages of incorporation were observed (Supplementary Fig. 1e)[16]. Furthermore, we have recently reported that stealthin C can be formed nonenzymatically in vivo and in vitro[20]. Therefore, we posit that stealthin C is not an authentic intermediate in kinamycin biosynthesis, and instead, the N–N bond is formed independently and later added to the polyketide scaffold in a convergent biosynthetic process. Not only do we observe the intact incorporation of the preformed N–N bond into kinamycin, we observe the incorporation of nitrite solely into the proximal nitrogen in the diazo functionality, which precludes a pathway involving late-stage diazotization. This observation also eliminates the possibility that free hydrazine is involved in the biosynthesis of kinamycin, although the exact N–N bond-containing intermediate transferred to the polyketide scaffold and the enzymes involved in this process remain to be elucidated.

Collectively, our results show that in the biosyntheses of the structurally highly diverse compounds fosfazinomycin and kinamycin, a hydrazine building block is channeled into the biosynthetic pathways by a common strategy. The N–N bond is first fashioned by conversion of aspartic acid to hydrazinosuccinic acid (**3**) in a process that involves nitrite but that has yet to be fully understood. Then a set of four conserved enzymes transfers the hydrazine moiety onto the side chain of glutamic acid before its final installation in the mature natural product (Fig. 6). This strategy is vastly different than in the biosyntheses of other N–N bond-containing natural products for which the N–N bond is formed directly on the scaffold of the final product. Thus, this study brings to light an unexpected pathway for the incorporation of N–N bonds in natural products.

## Methods

**General methods.** All reagents and materials for the fosfazinomycin and kinamycin biosynthetic studies were purchased from Sigma-Aldrich or Fisher Scientific unless otherwise noted. DNA oligonucleotides were obtained from Integrated DNA Technologies (IDT) and Sigma-Aldrich. Enzymes used in cloning were procured from New England Biosciences. DNA sequencing was performed by ACGT Inc., the Roy J. Carver Biotechnology Center (University of Illinois at Urbana-Champaign), or Eton Bioscience Inc. (Charleston, MA). NMR experiments were carried out on an Agilent 600 MHz with a OneNMR broadband probe, a Varian Inova-500 (500 MHz, 125 MHz), or a JOEL 400 (400 MHz, 100 MHz, 41 MHz), and the data were analyzed with MestreNova software. Gene cluster diagrams were constructed with the online Gene Graphics tool[26]. Genome sequencing of *Streptomyces murayamaensis* ATCC 21414 was performed by Era7 Bioinformatics (Granada, Spain) using Illumina MiSeq reads of two 280 bp paired-end libraries[27,28]. Annotations were carried out using Era7's BG7 tool. Sequence alignments were performed with Geneious. The relevant part of the *kin* gene cluster from *S. murayamaensis* ATCC 21414 has been deposited into NCBI (Accession # MH703729). The accession numbers for the *kin* gene sequences are in Supplementary Table 1.

**Isotope labeling experiments.** *Streptomyces* sp. NRRL S-149 was first cultivated on R2A agar for 2 d at 30 °C. A single colony was then picked and used to inoculate 5 mL of ATCC 172. After 3 d on a roller drum at 30 °C, 1 mL of the culture in ATCC 172 was used to inoculate a second seed culture in 25 mL of modified R2A medium (3.8 mM $^{15}NH_4Cl$, [$^{15}N$, 99%, Cambridge Isotope Laboratories], 0.05% soluble potato starch, 2.8 mM glucose, 2.7 mM sodium pyruvate, 0.9 mM potassium phosphate dibasic, and 0.2 mM magnesium sulfate heptahydrate) in a 125 mL flask. After 3 d of shaking at 200 rpm at 30 °C, 4 mL of the seed culture was used to inoculate a production culture consisting of 100 mL of modified R2AS (same as modified R2A above, supplemented with 100 μM $^{15}N$-aspartic acid [$^{15}N$, 99%, Cambridge Isotope Laboratories], 40 mM sodium succinate, and 0.5% Balch's vitamins) in a 500 mL baffled flask. The production culture was cultivated for 38 h at 30 °C with shaking at 200 rpm. Then 2 mM of acetylhydrazine, N-hydroxyaspartic acid (**2**), hydrazinosuccinic acid (**3**), $NaNO_2$, or glutamylhydrazine (**6**) was added to the culture. Compound **6** was prepared synthetically (Supplementary Methods), and **2** and **3** were obtained enzymatically from fumaric acid and either hydroxylamine or hydrazine using ammonia-aspartate lyase (AspB) from *Bacillus* sp YM55-1 using an adaptation of a previously reported method (50 μM AspB was incubated with 20 mM fumarate, 20 mM of either hydroxylamine or hydrazine in 50 mM $NaH_2PO_4$, pH = 7.7 for 8 h at ambient temperature.)[29]. The *Streptomyces* sp. NRRL S-149 production culture was then incubated for an additional 8 h at 30 °C with shaking at 200 rpm. The spent medium was concentrated 10X under reduced pressure and reconstituted in 80% MeOH. Precipitate was removed by centrifugation, and the supernatant was dried under reduced pressure and dissolved in 5 mL of $H_2O$. Solid phase extraction was performed with 300 mg of Oasis HLB resin (Waters). After the sample was loaded onto the resin, the resin was washed with 5% MeOH in $H_2O$, and fosfazinomycin A was eluted with 50% MeOH. LC-HRMS and LC-HR-MS/MS analyses in conjunction with the labeling experiments in *Streptomcyes* sp. NRRL S-149 were performed on a Thermo Q-Exactive Hybrid Quadrupole-Orbitrap Mass Spectrometer coupled to a Dionex Ultimate 3000 series HPLC system. An Xbridge $C_{18}$ column (4.6 × 250 mm, 5 μ, Waters) was used with mobile phase A ($H_2O$ with 0.1% formic acid) and mobile phase B (acetonitrile with 0.1% formic acid) at a flow rate of 0.5 mL min⁻¹. The chromatographic method consisted of a linear gradient of 5% B to 7.5% B in 10 min, a linear gradient from 7.5% B to 95% B in 7 min, and a linear gradient from 95% B to 5% B in 11 min. Analyses were performed in positive mode. Data were analyzed and processed with Thermo Xcalibur 3.0.63 software.

*S. murayamaensis* ATCC 21414 was cultivated on mannitol-soy agar for 5 d until sporulation. The spores were inoculated into 50 mL of ISP2 medium for 2 d with shaking at 220 rpm at 30 °C. Then, 1 mL of seed culture was transferred to 100 mL of production medium (glycerol 3%, $(NH_4)_2SO_4$ 0.1%, $K_2HPO_4$·$3H_2O$ 0.1%, $CaCO_3$ 0.1%, $MgSO_4$·$7H_2O$ 0.01%, and $FeSO_4$·$7H_2O$ 0.01%) containing either $^{15}N$-sodium nitrite [$^{15}N$, 98%, Cambridge Isotope Laboratories], $^{15}N$-calcium nitrate [$^{15}N$, 98%, Cambridge Isotope Laboratories], $^{15}N_2$-hydrazine sulfate [$^{15}N$, 98%, Cambridge Isotope Laboratories], or $^{15}N_2$-acetylhydrazine [prepared as described in Supplementary Methods] in a final concentration of 1 mM. $^{15}N_2$-**6** was added to the fermentation culture to a final concentration of 0.2 mM. The fermentation proceeded for 3 d, after which the cultures were centrifuged to remove the cells. The supernatants were extracted twice with ethyl acetate containing 1% acetic acid. The samples were concentrated, redissolved in methanol, and analyzed by LC-HRMS using an Agilent 1200 series LC system coupled to an Agilent qTOF 6530 mass spectrometer with a Hypersil Gold aq $C_{18}$ column (3 × 150 mm, 3 μ) using mobile phase C ($H_2O$ with 0.1% formic acid) and mobile phase D (acetonitrile with 0.1% formic acid) at a flow rate of 0.3 mL min⁻¹. The method consisted of a linear gradient from 5% D to 95% D in 15 min, an isocratic hold at 95% D for 5 min, a linear gradient from 95% D to 5% D in 4 min, and a final isocratic hold at 5% D for 5 min. Analyses were performed in positive ion mode, and the extracted ion chromatograms were generated using Agilent Chemstation software with a 5 ppm mass window.

For preparation of $^{15}N$-labeled kinamycin for NMR analysis, organic extracts from 500 mL of fermentation culture were prepared as described above. Kinamycin D was isolated from the extracts using a ThermoFisher Dionex Ultimate 3000 HPLC system with a Kromasil 100 $C_{18}$ column (10 × 150 mm, 5 μ) at a flow rate of 3 mL min⁻¹. The method consisted of a linear gradient from 5% D to 95% D in 30 min, an isocratic hold at 95% D for 10 min, and a linear gradient from 95 to 5% D in 8 min. Kinamycin D eluted at 23–25 min. The collected fractions were lyophilized and redissolved in $d_6$-DMSO for NMR analysis.

**Cloning.** Genomic DNA was isolated from *Streptomyces* sp. XY332, *Streptomyces* sp. WM6372, and *Streptomyces murayamaensis* ATCC 21414 using an Ultraclean Microbial DNA Isolation Kit (Mo Bio) following the manufacturer's instructions. From purified gDNA, the genes encoding FzmN (*Streptomyces* sp. XY332), FzmO (*Streptomyces* sp. WM6372), and FzmA (*Streptomyces* sp. XY332) were amplified using the primers listed in Supplementary Table 2 and Phusion polymerase. The PCR products were then purified using the QIAquick PCR Purification Kit (Qiagen). pET15b was linearized using NdeI, amplified by PCR with Phusion polymerase (primers listed in Supplementary Table 2), and treated with DpnI. *fzmN* and *fzmO* were ligated into the pET15b backbone, and *fzmA* was ligated into pET23b (digested with NdeI and XhoI) using a Gibson Assembly kit from New

England Biosciences. *E. coli* DH5α was used for transformation and plasmid production. The fidelity of the insertion was verified by DNA sequencing.

pET28a and the genes encoding KinK, KinL, and KinM were amplified using the primers listed in Supplementary Table 2. The linear vector and the insert were ligated using a Gibson Assembly Kit from New England Biosciences. The insertions were verified by DNA sequencing.

The codon-optimized gene encoding AspB from *Bacillus* sp. YM55-1 was ordered from IDT as a gBlock. Following PCR amplification with Q5 polymerase (primers and synthetic gene in Supplementary Table 2), *aspB* was ligated into pET15b (digested with NdeI and HindIII) using the HiFi Assembly Mix from New England Biosciences. The fidelity of the insertion was verified by DNA sequencing.

**Protein purification and refolding**. For FzmN, FzmA, or AspB, the appropriate vector was used to transform *E. coli* BL21(DE3). Transformants were cultivated for 12 h at 37 °C in LB with 100 µg mL$^{-1}$ ampicillin, and 40 mL of this culture was then used to inoculate 4 L of LB with 100 µg mL$^{-1}$ ampicillin. Cultivation continued at 37 °C with shaking at 220 rpm, until the OD$_{600}$ reached between 0.5 and 0.7, and IPTG (Gold Biotechnology) was added to 100 µM. The culture was grown for an additional 16 h at 18 °C with shaking at 220 rpm. The cells were then harvested by centrifugation and resuspended in lysis buffer (50 mM NaH$_2$PO$_4$, 300 mM NaCl, 10 mM imidazole, 10% glycerol) supplemented with 1 mg mL$^{-1}$ lysozyme (Gold Biotechnology) and 20 U mL$^{-1}$ DNase I. The cell suspension was then incubated with mild agitation for 20 min at 4 °C. Cell lysis was achieved by passaging the suspension twice through a French pressure cell. Insoluble cellular debris was removed by centrifugation for 45 min at 30,600× *g*. The supernatant was then loaded onto 5 mL of Ni-NTA resin pre-equilibrated with lysis buffer for 15 min with mild agitation. The resin was then washed with 40 mL of wash buffer (50 mM NaH$_2$PO$_4$, 300 mM NaCl, 20 mM imidazole, 10% glycerol), and the protein was eluted from the resin with 20 mL of elution buffer (50 mM NaH$_2$PO$_4$, 300 mM NaCl, 300 mM imidazole, 10% glycerol). The eluate was then concentrated to a volume of 2.5 mL using an Amicon spin filter with a 30 kDa molecular weight cutoff. Excess imidazole was removed using a PD-10 column (GE Healthcare) with 3.5 mL storage buffer (50 mM NaH$_2$PO$_4$, 300 mM NaCl, 10% glycerol). FzmO was expressed using the same methodology as for FzmN or FzmA. After cell lysis, the insoluble fraction was resuspended in pellet wash buffer (50 mM NaH$_2$PO$_4$, 300 mM NaCl, 0.10% Triton X-100, pH = 7.5) and passed through a 20 G syringe needle several times. Again the insoluble fraction was isolated from the supernatant via centrifugation, and the pellet was resuspended in denaturing buffer (50 mM NaH$_2$PO$_4$, 300 mM NaCl, 6 M guanidine HCl, 10 mM imidazole, 10% glycerol, pH = 7.5) and incubated at 56 °C for 15 min. After centrifugation, the supernatant was applied to Ni-NTA agarose resin (Qiagen). Nickel affinity purification proceeded similarly as with FzmN, except all buffers also contained 6 M guanidine hydrochloride.

Protein refolding conditions for FzmO were then screened using a platform adapted from another study[17]. Briefly, 100 µl of 216 different buffers (Supplementary Table 3) was placed into three clear round bottom 96-well plates (Corning). Each plate also contained 12 positive control wells, each filled with 91 µg bovine serum albumin in 110 µL of denaturing storage buffer (50 mM NaH$_2$PO$_4$, 300 mM NaCl, 6 M guanidine HCl, 10 mM imidazole, 10% glycerol, pH = 7.5); 12 negative control wells containing 91 µg insoluble FzmO in 110 µl of storage buffer (without guanidine hydrochloride) were also included in each plate. Then, 10 µL of a stock solution of 1 mg mL$^{-1}$ FzmO in denaturing storage buffer was diluted into each of the 216 buffers in the 96-well plates. The plates were incubated for 12 h at room temperature with gentle agitation. The turbidity of each well was then measured with a Tecan Infinite 200 Pro plate reader, reading absorbance at λ = 410 nm and a bandwidth of 9 nm using multiple circle-filled 3 × 3 reads and 10 flashes per read. Each plate was prepared and analyzed in triplicate on different days. Hits were defined as conditions that gave rise to absorbance readings within three standard deviations of the mean absorbance of the positive control wells.

pET28a_*kinN* and pET28a_*kinM* were used to transform chemically competent *E. coli* BL21 (DE3) cells, and the resulting transformants were cultivated overnight in 12 mL of starter cultures at 37 °C in LB with 50 µg mL$^{-1}$ of kanamycin. The next day, 10 mL of this starter culture was used to inoculate 1 L of LB medium containing 50 µg mL$^{-1}$ kanamycin. The cells were grown at 37 °C with shaking at 180 rpm until they reached an OD$_{600}$ of ~0.4–0.6. Then, 200 µM of IPTG was added, and the culture was grown for an additional 16 h at 15 °C. The cells were harvested, resuspended in lysis buffer (20 mM HEPES pH 8.0, 500 mM NaCl), and passed through a cell disruptor. The lysate was clarified via centrifugation, and the resulting supernatant was incubated with Ni-NTA resin for 1 h at 4 °C on a nutating mixer. The subsequent purification steps were analogous to the protocol for purifying FzmN with the exception of the compositions of the wash buffer (50 mM HEPES pH 8.0, 500 mM NaCl, 20 mM imidazole), elution buffer (50 mM HEPES pH 8.0, 500 mM NaCl, 200 mM imidazole), and storage buffer (50 mM HEPES pH 8.0, 50 mM NaCl, 10 mM MgCl$_2$, 5% glycerol).

**Enzyme assays**. To test whether FzmN catalyzes the canonical glutamine synthetase reaction, 5 mM glutamic acid, 6 mM NH$_4$Cl, 50 µM FzmN, 6 mM ATP, and 6 mM MgCl$_2$ were incubated in 50 mM NaH$_2$PO$_4$, pH = 7.7 for 5 h at room temperature. The reaction mixture was then treated with Chelex 100 resin for 15 min, and the enzyme was removed by filtration through an Amicon Ultra 30 K

centrifugal filter (EMD Millipore) before NMR analysis. To test whether FzmN catalyzes the formation of glutamylacetylhydrazine (**5**), 5 mM glutamic acid, 6 mM acetylhydrazine, 34 µM FzmN, 6 mM ATP, and 6 mM MgCl$_2$ were incubated in 50 mM NaH$_2$PO$_4$, pH = 7.7 for 8 h at room temperature before Chelex treatment, enzyme removal, and NMR analysis.

To generate Michaelis–Menten curves for FzmN with NH$_4$Cl and glutamic acid, a coupled assay system consisting of pyruvate kinase and lactate dehydrogenase was employed and NADH consumption was measured at 340 nm with a Cary 4000 spectrometer. The concentration of NH$_4$Cl was varied (4 mM to 100 mM) while maintaining a constant concentration of FzmN (1.77 µM). When the concentration of NH$_4$Cl was below 10 mM, 100 mM glutamic acid, 36 mM MgCl$_2$, 3.75 mM ATP, 1.25 mM phosphoenolpyruvic acid, 200 µM NADH, 50 U mL$^{-1}$ pyruvate kinase, and 75 U mL$^{-1}$ lactate dehydrogenase were mixed in 50 mM Tris-HCl, pH = 7.7, and the reaction was initiated by adding NH$_4$Cl. When the concentration of NH$_4$Cl used was 10 mM or higher, the same reaction setup was used but with decreased concentrations of pyruvate kinase (30 U mL$^{-1}$) and lactate dehydrogenase (45 U mL$^{-1}$). Attempts to generate kinetic parameters for FzmN towards acetylhydrazine and glutamic acid were executed similarly using the following setup. The concentration of acetylhydrazine was varied (10 µM to 200 µM) while maintaining a constant concentration of FzmN (0.106 µM). Glutamic acid (100 mM), 36 mM MgCl$_2$, 5 mM ATP, 2 mM phosphoenolpyruvic acid, 200 µM NADH, 40 U mL$^{-1}$ pyruvate kinase, and 60 U mL$^{-1}$ lactate dehydrogenase were mixed in 50 mM Tris-HCl, pH = 7.7, and the reaction was initiated by adding acetylhydrazine. Each experiment was performed in triplicate, and the standard deviation was calculated.

In order to test if FzmO is able to deacetylate glutamylacetylhydrazine (**5**), 50 µL was withdrawn from a well in the 96-well plate that was defined as a hit (~10 µM FzmO in 50 mM HEPES, 100 mM KCl, 20% glycerol, 800 mM arginine, pH = 9.0) in the refolding screen. Then, compound **5** was added to a final concentration of 1 mM, and the reaction was allowed to proceed for 11 h at room temperature. The protein was then removed by an Amicon Ultra 10 K filter, and the reaction mixture was derivatized by adding Fmoc-Cl to the reaction mixture (20 mM final). This mixture was incubated for 20 min with agitation, and 1-aminoadamantane was added to 50 mM, after which incubation continued for another 20 min. A negative control was also run with the same reaction mixture but without any FzmO. After derivatization, the reactions were analyzed by LC-MS using an Agilent 1200 series LC system coupled to an Agilent G1956D single quadrupole mass spectrometer with an Xbridge C$_{18}$ column (4.6 × 250 mm, 5 µ) using mobile phase A (H$_2$O) and mobile phase B (acetonitrile) at a flow rate of 1 mL min$^{-1}$. The method consisted of an isocratic hold at 5% B for 1 min, a linear gradient from 5% B to 95% B in 29 min, an isocratic hold at 95% B for 10 min, a linear gradient from 95% B to 5% B in 4 min, and a final isocratic hold at 5% B for 21 min. A positive control was also run wherein 500 µM Fmoc-derivatized glutamylhydrazine was spiked into the reaction mixture. Extracted ion chromatograms were generated using Agilent Chemstation software.

To test the activity of FzmA, 5 mM compound **6**, 5 mM arginine, 6 mM MgCl$_2$, 6 mM ATP, 6 mM thermostable inorganic phosphatase (New England Biosciences), and 21 µM FzmA were incubated in 50 mM NaH$_2$PO$_4$, pH = 7.7 at ambient temperature for 21 h. A negative control was also run using the same conditions as above except that FzmA was withheld from the reaction mixture. After passage through a spin filter to remove enzyme, the reaction mixture was subjected to Fmoc-Cl derivatization as described above before analysis by LC-MS with an Agilent 1200 series LC system coupled to an Agilent G1956B single quadrupole mass spectrometer using a Grace Vydac C$_{18}$ column (4.6 × 250 mm, 5 µ). The method used mobile phase A (H$_2$O) and mobile phase B (acetonitrile) at a flow rate of 0.4 mL min$^{-1}$ and consisted of an isocratic hold at 20% B for 2 min, a linear gradient from 20 to 100% B in 30 min, an isocratic hold at 100% B for 2 min, a linear gradient from 100 to 20% B in 5 min, and an isocratic hold at 20% B for 5 min. The analysis was performed in negative mode with selected ion monitoring for Fmoc-derivatized glutamic acid ([M-H]$^-$ = 368).

To test the activity of KinL, *E. coli* BL21 (DE3) harboring pET28a_*kinL* was grown to OD$_{600}$ ~0.5 before adding 0.2 mM IPTG, 1 mM glutamic acid, and 1 mM acetylhydrazine or hydrazine. A parallel control with BL21 (DE3) harboring an empty pET28a vector was also prepared. The cells were allowed to grow overnight at 15 °C. The next day, the cultures were centrifuged, and 0.5 mL of the supernatant was added to an equivalent volume of acetone. The samples were centrifuged to remove precipitates and analyzed by LC-HRMS using an Agilent 1200 series LC system coupled to an Agilent qTOF 6530 mass spectrometer. The column used was a Cogent Diamond Hydride column (3 × 150 mm, 4 µ) using mobile phase A (H$_2$O with 0.1% formic acid) and mobile phase B (acetonitrile with 0.1% formic acid) at a flow rate of 0.5 mL min$^{-1}$. The method consisted of an isocratic hold at 10% A for 1 min, a linear gradient from 10% A to 70% A in 19 min, an isocratic hold at 70% A for 1 min, a linear gradient from 70 to 10% A in 4 min, and an isocratic hold at 10% A for 4 min.

To test the activity of KinN, 2 mM hydrazinosuccinic acid (**3**), 2 mM acetyl-CoA, and 20 µM KinN were incubated in 50 mM potassium phosphate buffer pH 8.0 and 10 mM MgCl$_2$ (200 µL total volume). Two negative controls lacking KinN or acetyl-CoA were prepared in parallel. After 1 h, the assay mixtures were frozen and lyophilized to dryness before resuspension in D$_2$O and analysis by $^1$H NMR spectroscopy. To test the activity of KinM, KinM was added directly to the NMR sample to a final concentration of 10 µM, and the reaction was monitored by $^1$H NMR spectroscopy over the course of 1 h.

For LC-HRMS analysis, the same assays were performed in final volumes of 50 μL. The assay mixtures were then analyzed by LC-HRMS in an analogous manner to the procedure for detecting isotopically labeled kinamycin D with the exception that the column used was an Acclaim™ Polar Advantage $C_{18}$ column (ThermoFisher Scientific) (2.1 × 150 mm, 3 μ) and negative ion mode was used. To confirm the consumption of **3** in the activity assay of KinNM, the liquid chromatographic method for detecting **5** was used.

**Chemical synthesis of substrates and standards**. The procedures for the chemical synthesis of the various substrates and standards are described in the Supplementary Methods (Supplementary Fig. 9, 14, 19, and 25) and spectroscopic data are presented in Supplementary Fig. 10-13, 15-18, 20-24, and 26-27.

## Data availability
Data generated or analyzed during this study are included in this published article (and its supplementary information files). Raw data for mass and NMR spectra are available from the corresponding authors on request. The sequences of the genes in the *kin* cluster from *S. murayamaensis* ATCC 21414 that encode for proteins discussed in this study have been deposited to Genbank (MH703729).

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

## Acknowledgements
This work was supported by the National Institutes of Health (GM P01 GM077596 to W.A.V. and GM DP2 GM105434 to E.P.B.), a Cottrell Scholar Award (to E.P.B.), a Camille Dreyfus Teacher-Scholar Award (to E.P.B.), and Harvard University (to E.P.B.). We thank Dr. Zhong Li of the Metabolomics Laboratory of the Roy J. Carver Biotechnology Center (UIUC) for acquiring HRMS and MS/MS spectra, and Nicholas Lue (Harvard University) for assisting with chemical synthesis. A subset of NMR spectra was acquired on a 600 MHz instrument purchased with support from NIH Grant S10 RR028833.

## Author contributions
Experiments were designed by W.A.v.d.D., E.P.B., K-K.A.W, Z.H., T.L.N. and P.W. Experiments were performed by K-K.A.W., T.L.N., P.W. and Z.H. The manuscript was written by K-K.A.W., T.L.N., E.P.B. and W.A.v.d.D. The research was supervised by E.P.B. and W.A.v.d.D.

## Additional information

**Competing interests:** The authors declare no competing interests.

