## [Peer Review File · Nature Communications]

Reviewers' comments:

Reviewer #1 (Remarks to the Author):

The article reviewed presents work towards the characterization of two groups of homologous gene products, putatively thought to be involved in the formation of the N-N bonds in the structurally unrelated compounds fosfazinomycin and kinamycin. This represents an expansion and advancement on work the group had previously published in 2016 (Chem. Sci., 2016, 7, 5219-5223).

In the case of fosfazinomycin, the authors implemented a clever method utilizing unlabeled synthetic intermediates and introducing them via feeding experiments into globally ¹⁵N labeled media to observe incorporation. This removed the need for synthesis of expensive and challenging putative ¹⁵N-intermediates and still allowed for a way of observing incorporation via LCHRMS analysis. The authors were successful in showing the incorporation of several key intermediates into the structure of fosfazinomycin through these feeding experiments, enabling a better understanding of how N-N bond formation was being facilitated, and the source of the nitrogen atoms within the N-N bond. The authors went on to further characterize the enzyme activity in vitro, leaving no doubt of the function of the enzymes, and mapping out a plausible pathway to N-N bond formation. Some limited kinetic data was also presented.

The authors then probed kinamycin biosynthesis performing feeding experiments using ¹⁵N-nitrite. Labeled kinamycin was isolated and found to be labeled at the proximal nitrogen of the diazo-group suggesting a different mode of incorporation than was previously believed. Again, the authors go on to demonstrate the activity of several of the enzymes in vitro.

This represents an important step in understanding the biosynthetic mechanism by which N-N bonds are formed in natural products. Of specific interest is identification of a new pathway for the incorporation of N-N bonds into natural products in which hydrazine is synthesized separately from the natural product core and is then coupled using a glutamyl carrier system.

A primary concern is that this study represents an interesting albeit incomplete story. Specifically, the work does not provide a full understanding of how the hydrazine is initially formed, and how the final incorporation step occurs, especially in the case of kinamycin. The work does provide evidence that these are indeed the enzymes responsible for shuttling hydrazine within the respective biosynthetic pathways. Thus, although the work described is well performed and of high impact, but without the initial step or the final step, this represents a fractured story.

Major Concerns in the Current Manuscript

The authors discuss their opinion regarding how the hydrazine is transferred to arginine for incorporation into fosfazinomycin. Unfortunately, a detailed discussion of how they believe hydrazine is transferred to the kinamycin core is absent. This proposal should be included to strengthen the paper in the absence of in vitro data. The structure that is drawn in Figure 6 (the hydroxy-precursor of kinamycin) may not be the best structure to include. The authors may want to consider the isolated compound kinobscurinone (reference 13 in the article) as a more likely intermediate for the final coupling of hydrazine towards formation of the diazo functionality. Could a coupling with an aldehyde formed during C-C bond cleavage of dehydrabelomycin (mentioned in reference 17 of the article), in similar fashion to jadomycin biosynthesis, giving a hydrozone intermediate, which cyclizes and oxidizes to the diazo group be possible?

Another concern regards the claim by the authors that there is no spontaneous free hydrazine incorporation into kinamycin. The authors should discuss why they believe no labelling occurs at 0.2 mM hydrazine feeding while labelling is observed in 1 mM hydrazine feeding experiments. It appears that the promiscuous activity of KinL may play a role. The hydrazine KinL in vivo experiments should be conducted at both 0.2 mM and 1 mM hydrazine concentrations.

General Suggestions

Page 2, main text, line 16 and 18: the authors mention cytotoxic activity. Please consider including what specifically these compounds have cytotoxicity towards.

Page 3, line 5: the authors in this case have referred to the compound kutzneride (I believe this is the correct spelling), but in figure 1 they spell it "kutzernide", please make sure this spelling is

correct throughout the manuscript.

Page 3, line 11 and 12: "For instance, during the biosynthesis of cremeomycin, a flavin-dependent monooxygenase, CreE,..." should read "For instance, during the biosynthesis of cremeomycin, CreE, a flavin-dependent monooxygenase...". As it currently reads it sounds like cremeomycin is a flavin-dependent monooxygenase.

Page 3, line 19, 20 and 21: Please consider revising the awkward sentence, "Moreover, we showed that aspartic acid is incorporated into at least one of the nitrogen atoms in the phosphonohydrazide moiety of fosfazinomycin."

Aspartic acid is not incorporated into the nitrogen atom. This should be changed to something closer to "aspartic acid is responsible for the incorporation of at least one of the nitrogen atoms".

Page 4, line 19 and 20: "from the biosynthetic routes to kutzneride and cremeomycin in which the N-N bond is formed directly on a biosynthetic intermediate" should read "from the biosynthetic routes to kutzneride and cremeomycin in which the N-N bonds are formed directly on biosynthetic intermediates".

Page 6, line 1: please consider replacing "We next fed 2 at natural abundance" with "We next fed unlabeled 2", the current wording is confusing.

Page 6, line 9: please fix the spelling of medium (currently spelled medum).

Page 6, line 10: remove the word "added".

Page 6, line 22: the use of competent is awkward, please consider removing or changing.

Page 8, line 4: remove the word thus.

Page 8, line 14: please consider rewording the sentence starting with "In this study" (or incorporating it into the previous sentence). The use of this was confusing as it initially made me think the sentence was about the article under review. Instead the authors were referring to another study which was not immediately apparent.

Page 9, line 6: please consider replacing "which we have labeled" to "which were labeled".

Page 9: In general, throughout the paper try to reduce the use of "we", it is understood that the authors did the work and it is not required (six instances of the use "we" on page 9).

Page 10, line 12: please change catalyzes to catalyzed.

Page 11, line 14: remove the word different.

Page 12, line 2: replace "will" with "would"

Page 18, General methods, last sentence: please italicize *S. murayamaensis*.

Page 19, line 4: please superscript the second ¹⁵N in the line (to be consistent with page 18 when describing the labeled NH₄Cl used).

Page 19, line 7: please italicize the N in N-hydroxyaspartic acid.

Page 20, lines 8, 9, and 10: please be consistent with superscript ¹⁵N throughout the paper.

Page 26, line 20: "After a passage a spin filter" should read "After a passage through a spin filter".

Page 27, line 7: the authors have written pET28a-kinL, throughout the rest of the paper this (and other plasmid constructs) has been written with an underscore, please be consistent throughout the paper.

Page 34, figure 3, heading line 8: please change "⁵N-nitrite" to "¹⁵N-nitrite".

SI Table of Contents: please include page numbers next to headings.

Supplementary figure 1: Please include heading and figure on the same page.

Reviewer #2 (Remarks to the Author):

The manuscript "Glutamic acid is a carrier for hydrazine during the biosynthesis of fosfazinomycin and kanamycin" offers a fascinating study on an unusual class of diazo-group containing natural products. Unifying what is known about the biosynthesis of two molecules, fosfazinomycin A and kinamycin D, which are structurally distinct but share a diazo functionality, the authors piece

together the fundamental pathway of nitrogen-nitrogen incorporation. The manuscript follows a logical experimental path and incorporates the clever use of isotope labeling with a combination of NMR and MS methodology. The reader is left with a strong understanding of how nature incorporates N-N bonds into highly disparate pathways. Overall, the manuscript may be suitable for the Nature Communications audience, but a few issues should be addressed:

1. The manuscript discusses homologies of the proteins in question, both between the pathways and with putative activity characterizations. However there are no figures comparing sequences or indicating how activities are predicted. These should be added to the SI.
2. Numbering of the compounds is confusing and irregular. Compound 6 is made as both an unlabeled and ^{15}N -isotope enriched forms. The numbering schemes to differentiate these compounds in the text and the SI should be corrected. Either the isotope enriched form requires a different number, or it should always be preceded by a prefix that clarifies the compound, such as " ^{15}N -6". In addition, ^{15}N acetylhydrazine has no number in the text and SI, which should be corrected.
3. Given the number and isotope identity of compounds synthetically prepared, spectroscopic data should be provided in the SI. At a minimum, ^1H -NMR spectra of final compounds should be shown, as is the norm for chemical journals.
4. Lack of full reconstitution of the FzmA activity is a disappointment and brings into question some of the comparisons with asparagine synthetase. If indeed the activity is as claimed, the authors should be able to demonstrate adenylation activity of the enzyme by following ATP consumption and hydrolysis of the aminoacyl-AMP intermediate. If this assay wasn't performed, why not?
5. There is no proposal in the manuscript for incorporation of 6 into kanamycin, except to find that hydrazine alone does not carry out this reaction. This missing link is another disappointment in the manuscript that leaves more questions than answers.
6. Figure 3 caption, line 8. Should be " ^{15}N -nitrite"

Overall this is a strong manuscript from a team of world-class natural product scientists. The experiments are bold and thoughtful. The findings are impressive. The end was a little weak and left me wanting more. Still, I thoroughly enjoyed reading it.

Reviewer #3 (Remarks to the Author):

This manuscript describes detailed investigations of the origins of the nitrogen atoms of the N-N bond present in the natural products fosfazinomycin and kanamycin. Results are also presented that provide important details about the pathways employed for the creation of the N-N bond. The investigators make use of elegant precursor incorporation experiments as well as in vitro enzymology to tease out some of the complexities of these pathways. The studies reveal that, for both fosfazinomycin and kanamycin, one of the nitrogen atoms in the N-N bond originates from nitrous acid. Additional studies show that both pathways involve the intermediacy of acetylhydrazine and glutamylhydrazine as carriers of a hydrazine moiety. Furthermore, the results from experiments with ^{15}N -labeled precursors preclude the intermediacy of free hydrazine in either of these pathways. These findings reveal a biosynthetic strategy that is completely different from the biosynthesis of some N-N bond containing natural products where the N-N bond is formed from two nitrogen atoms already located on the scaffold of the final product. The results described here provide highly novel and important insights regarding the biosynthesis of some N-N bond containing natural products. They will be of great interest to investigators studying natural product biosynthesis. The experimental and supplementary data included with the manuscript provide compelling evidence in support of the conclusions drawn by the authors.

Reviewer comments are in regular font and our responses are in *italic font*.

Reviewer #1 (Remarks to the Author):

The article reviewed presents work towards the characterization of two groups of homologous gene products, putatively thought to be involved in the formation of the N-N bonds in the structurally unrelated compounds fosfazinomycin and kinamycin. This represents an expansion and advancement on work the group had previously published in 2016 (Chem. Sci., 2016, 7, 5219-5223). In the case of fosfazinomycin, the authors implemented a clever method utilizing unlabeled synthetic intermediates and introducing them via feeding experiments into globally ¹⁵N labeled media to observe incorporation. This removed the need for synthesis of expensive and challenging putative ¹⁵N-intermediates and still allowed for a way of observing incorporation via LCHRMS analysis. The authors were successful in showing the incorporation of several key intermediates into the structure of fosfazinomycin through these feeding experiments, enabling a better understanding of how N-N bond formation was being facilitated, and the source of the nitrogen atoms within the N-N bond. The authors went on to further characterize the enzyme activity in vitro, leaving no doubt of the function of the enzymes, and mapping out a plausible pathway to N-N bond formation. Some limited kinetic data was also presented.

The authors then probed kinamycin biosynthesis performing feeding experiments using ¹⁵N-nitrite. Labeled kinamycin was isolated and found to be labeled at the proximal nitrogen of the diazo-group suggesting a different mode of incorporation than was previously believed. Again, the authors go on to demonstrate the activity of several of the enzymes in vitro.

This represents an important step in understanding the biosynthetic mechanism by which N-N bonds are formed in natural products. Of specific interest is identification of a new pathway for the incorporation of N-N bonds into natural products in which hydrazine is synthesized separately from the natural product core and is then coupled using a glutamyl carrier system.

REPLY: We appreciate recognition of the novelty of this study and the important step this work represents.

A primary concern is that this study represents an interesting albeit incomplete story. Specifically, the work does not provide a full understanding of how the hydrazine is initially formed, and how the final incorporation step occurs, especially in the case of kinamycin. The work does provide evidence that these are indeed the enzymes responsible for shuttling hydrazine within the respective biosynthetic pathways. Thus, although the work described is well performed and of high impact, but without the initial step or the final step, this represents a fractured story.

REPLY: We agree that this study does not answer all questions, but as mentioned by all reviewers, this study convincingly shows an unexpected and novel way of introducing N-N bonds in multiple pathways to diverse structures. It provides the ground work for future studies.

Major Concerns in the Current Manuscript

The authors discuss their opinion regarding how the hydrazine is transferred to arginine for incorporation into fosfazinomycin. Unfortunately, a detailed discussion of how they believe hydrazine is transferred to the kinamycin core is absent. This proposal should be included to strengthen the paper in the absence of in vitro data. The structure that is drawn in Figure 6 (the hydroxy-precursor of kinamycin) may not be the best structure to include. The authors may want to consider the isolated compound kinobscurinone (reference 13 in the article) as a more likely intermediate for the final coupling of hydrazine towards formation of the diazo functionality.

Wilfred A. van der Donk
Investigator
Richard E. Heckert Professor of Chemistry

University of Illinois at Urbana-Champaign
Department of Chemistry
600 South Mathews Avenue, Box 38-5, Urbana, IL 61801

REPLY: As requested, we discuss proposals for how glutamylhydrazine could be incorporated into kinamycin (Supplementary Fig. 8). We propose that either: 1) the glutamyl hydrazide first forms an adduct with the polyketide and then a peptidase or amidase-like enzyme hydrolytically removes the amino acid, or 2) a glutamine amidotransferase-like enzyme catalyzes the hydrolysis of glutamylhydrazine to generate hydrazine in the enzyme active site, and then this enzyme or a separate partner enzyme transfers hydrazine to the polyketide scaffold. The enzymes that catalyze these transformations may or may not be encoded within the kin/lom gene clusters, but we have identified KinW, a glutamine-amidotransferase-like enzyme present in both gene clusters, as a potential candidate for this reaction. These biosynthetic hypotheses are now discussed in the revised text and figures. We also agree with the reviewer that kinobscurinone is a likely candidate coupling partner, and we have now incorporated this intermediate into Fig. 6 in the main text and Supplementary Fig. 8.

Could a coupling with an aldehyde formed during C-C bond cleavage of dehydrorabelomycin (mentioned in reference 17 of the article), in similar fashion to jadomycin biosynthesis, giving a hydrozone intermediate, which cyclizes and oxidizes to the diazo group be possible?

REPLY: We think this is a reasonable hypothesis and have incorporated this proposal into Supplementary Figure 8. We thank the reviewer for the suggestion.

Another concern regards the claim by the authors that there is no spontaneous free hydrazine incorporation into kinamycin. The authors should discuss why they believe no labelling occurs at 0.2 mM hydrazine feeding while labelling is observed in 1 mM hydrazine feeding experiments. It appears that the promiscuous activity of KinL may play a role.

REPLY: When we fed hydrazine to S. murayamaensis, the medium used was glycerol ammonium medium with ammonium concentration at 0.1% (or ~ 8 mM). We hypothesize that at 0.2 mM, the hydrazine is probably not effectively competing with ammonium and acetyl hydrazine for use by KinL.

The hydrazine KinL in vivo experiments should be conducted at both 0.2 mM and 1 mM hydrazine concentrations.

REPLY: We performed this experiment and observed similar levels of production of glutamylhydrazide (6) when 0.2 mM or 1 mM hydrazine were added to cultures of E. coli overexpressing KinL. We noted that the optical density of the final culture was lower when 1 mM hydrazine was used, suggesting that hydrazine is inhibiting cell growth and possibly protein overexpression levels. Potential explanations for the discrepancy between this result and the Streptomyces feeding experiment could be differences in (1) protein expression in the two organisms, (2) primary and secondary metabolism of hydrazine in the two organisms, and (3) composition of the fermentation medium. Overall, the data obtained from examining the activity of KinL in vivo indicate the possibility of promiscuous activity of KinL and potentially explain why free hydrazine is incorporated into kinamycin D in feeding experiments.

General Suggestions

Page 2, main text, line 16 and 18: the authors mention cytotoxic activity. Please consider including what specifically these compounds have cytotoxicity towards.

We added text explaining the type of cytotoxicity.

Page 3, line 5: the authors in this case have referred to the compound kutzneride (I believe this is the

correct spelling), but in figure 1 they spell it “kutzernide”, please make sure this spelling is correct throughout the manuscript.

We agree and thank the reviewer for pointing out the typo, which has now been corrected.

Page 3, line 11 and 12: “For instance, during the biosynthesis of cremeomycin, a flavin-dependent monooxygenase, CreE,...” should read “For instance, during the biosynthesis of cremeomycin, CreE, a flavin-dependent monooxygenase...”. As it currently reads it sounds like cremeomycin is a flavin-dependent monooxygenase.

We agree and made the suggested change.

Page 3, line 19, 20 and 21: Please consider revising the awkward sentence, “Moreover, we showed that aspartic acid is incorporated into at least one of the nitrogen atoms in the phosphonohydrazide moiety of fosfazinomycin.”

Aspartic acid is not incorporated into the nitrogen atom. This should be changed to something closer to “aspartic acid is responsible for the incorporation of at least one of the nitrogen atoms”.

We agree and rephrased this sentence to address the problem pointed out by the referee.

Page 4, line 19 and 20: “from the biosynthetic routes to kutzneride and cremeomycin in which the N-N bond is formed directly on a biosynthetic intermediate” should read “from the biosynthetic routes to kutzneride and cremeomycin in which the N-N bonds are formed directly on biosynthetic intermediates”.

We agree and made the suggested change.

Page 6, line 1: please consider replacing “We next fed 2 at natural abundance” with “We next fed unlabeled 2”, the current wording is confusing.

We agree and made the suggested change here and elsewhere.

Page 6, line 9: please fix the spelling of medium (currently spelled medum).

We appreciate spotting the typo and have corrected it.

Page 6, line 10: remove the word “added”.

Done

Page 6, line 22: the use of competent is awkward, please consider removing or changing.

Done

Page 8, line 4: remove the word thus.

Done

Page 8, line 14: please consider rewording the sentence starting with “In this study” (or incorporating it into the previous sentence). The use of this was confusing as it initially made me think the sentence was about the article under review. Instead the authors were referring to another study which was not immediately apparent.

We agree and made the suggested change.

Page 9, line 6: please consider replacing “which we have labeled” to “which were labeled”.
We agree and deleted “which we have”

Page 9: In general, throughout the paper try to reduce the use of “we”, it is understood that the authors did the work and it is not required (six instances of the use “we” on page 9).
Nature-Springer encourages the use of active rather than passive tense. We removed some of the use of “we” but generally tried to retain active tense as much as possible.

Page 10, line 12: please change catalyzes to catalyzed.
We agree and made the suggested change.

Page 11, line 14: remove the word different.
We agree and made the suggested change.

Page 12, line 2: replace “will” with “would”
We agree and made the suggested change.

Page 18, General methods, last sentence: please italicize *S. murayamaensis*.
We agree and made the suggested change.

Page 19, line 4: please superscript the second ¹⁵N in the line (to be consistent with page 18 when describing the labeled NH₄Cl used).
We agree and made the suggested change.

Page 19, line 7: please italicize the N in N-hydroxyaspartic acid.
We agree and made the suggested change.

Page 20, lines 8, 9, and 10: please be consistent with superscript ¹⁵N throughout the paper.
We agree and have inspected the paper for consistency and made revisions where needed.

Page 26, line 20: “After a passage a spin filter” should read “After a passage through a spin filter”.
We agree and made the suggested change.

Page 27, line 7: the authors have written pET28a-kinL, throughout the rest of the paper this (and other plasmid constructs) has been written with an underscore, please be consistent throughout the paper.
We agree and have inspected the paper for consistency and made revisions where needed.

Page 34, figure 3, heading line 8: please change “⁵N-nitrite” to “¹⁵N-nitrite”.
We appreciate the reviewer spotting this mistake and have corrected it.

SI Table of Contents: please include page numbers next to headings.
Done

Supplementary figure 1: Please include heading and figure on the same page.
Done

We greatly appreciate the detailed review of reviewer 1 that has very much improved the manuscript.

Reviewer #2 (Remarks to the Author):

The manuscript "Glutamic acid is a carrier for hydrazine during the biosynthesis of fosfazinomycin and kanamycin" offers a fascinating study on an unusual class of diazo-group containing natural products. Unifying what is known about the biosynthesis of two molecules, fosfazinomycin A and kinamycin D, which are structurally distinct but share a diazo functionality, the authors piece together the fundamental pathway of nitrogen-nitrogen incorporation. The manuscript follows a logical experimental path and incorporates the clever use of isotope labeling with a combination of NMR and MS methodology. The reader is left with a strong understanding of how nature incorporates N-N bonds into highly disparate pathways. Overall, the manuscript may be suitable for the Nature Communications audience, but a few issues should be addressed:

REPLY: We appreciate the reviewer's comments.

1. The manuscript discusses homologies of the proteins in question, both between the pathways and with putative activity characterizations. However there are no figures comparing sequences or indicating how activities are predicted. These should be added to the SI.

REPLY: We appreciate the suggestion and added the %identity between orthologous enzymes to Fig. 1 and added Table S1 with a more complete comparison.

2. Numbering of the compounds is confusing and irregular. Compound 6 is made as both an unlabeled and ¹⁵N-isotope enriched forms. The numbering schemes to differentiate these compounds in the text and the SI should be corrected. Either the isotope enriched form requires a different number, or it should always be preceded by a prefix that clarifies the compound, such as "¹⁵N-6". In addition, ¹⁵N-acetylhydrazine has no number in the text and SI, which should be corrected.

REPLY: We agree and have chosen to always use the prefix. However, we don't think a simple compound like acetylhydrazine should be replaced by a number. Readers usually will have an easier time following the discussion when we use acetylhydrazine than when we use a number, which needs to be looked up. Whereas for complex compounds, a number is preferred over a complex name, we feel that a simple compound is better called by its name rather than a number.

3. Given the number and isotope identity of compounds synthetically prepared, spectroscopic data should be provided in the SI. At a minimum, ¹H-NMR spectra of final compounds should be shown, as is the norm for chemical journals.

REPLY: We have added the requested spectra for all compounds in the new Supplementary Information.

4. Lack of full reconstitution of the FzmA activity is a disappointment and brings into question some of the comparisons with asparagine synthetase. If indeed the activity is as claimed, the authors should be able to demonstrate adenylation activity of the enzyme by following ATP consumption and hydrolysis of the aminoacyl-AMP intermediate. If this assay wasn't performed, why not?

REPLY: Yes, we did do these experiments but have not been able to observe adenylation. When FzmA is incubated with ATP, we see the formation of ADP and Pi instead; we also observe the analogous conversion of CTP, GTP, or UTP to their respective NDPs and Pi. We are not sure if this is real or NTPase activity from a contaminant. Hence, we prefer not to speculate about this activity at present.

Wilfred A. van der Donk
Investigator
Richard E. Heckert Professor of Chemistry

University of Illinois at Urbana-Champaign
Department of Chemistry
600 South Mathews Avenue, Box 38-5, Urbana, IL 61801

5. There is no proposal in the manuscript for incorporation of 6 into kanamycin, except to find that hydrazine alone does not carry out this reaction. This missing link is another disappointment in the manuscript that leaves more questions than answers.

REPLY: As requested and outlined in our response to comments from Reviewer 1, we now discuss proposals for how glutamylhydrazine could be incorporated into kinamycin in the main text and summarize these hypotheses in Supplementary Fig. 8. We propose that either the glutamyl hydrazide first forms an adduct with the polyketide and then a peptidase or amidase-like enzyme hydrolyzes the amino acid, or that a glutamine amidotransferase-like enzyme releases hydrolyzes the glutamyl hydrazide to generate hydrazine in its active site and then catalyzes the transfer to the polyketide scaffold. We also identify several potential polyketide partners for this reaction. Finally, we conducted a thorough comparative analyses of the kin and lom gene clusters to identify a conserved glutamine-amidotransferase-like enzyme (KinW) that might play a role in transfer.

6. Figure 3 caption, line 8. Should be "¹⁵N-nitrite"

REPLY: We appreciate the reviewer pointing out this mistake and have corrected it.

Overall this is a strong manuscript from a team of world-class natural product scientists. The experiments are bold and thoughtful. The findings are impressive. The end was a little weak and left me wanting more. Still, I thoroughly enjoyed reading it.

REPLY: We appreciate the kind words and hope that the reviewer likes the additional proposal of how glutamylhydrazine might be transferred to the polyketide scaffold.

Reviewer #3 (Remarks to the Author):

This manuscript describes detailed investigations of the origins of the nitrogen atoms of the N-N bond present in the natural products fosfazinomycin and kanamycin. Results are also presented that provide important details about the pathways employed for the creation of the N-N bond. The investigators make use of elegant precursor incorporation experiments as well as in vitro enzymology to tease out some of the complexities of these pathways. The studies reveal that, for both fosfazinomycin and kanamycin, one of the nitrogen atoms in the N-N bond originates from nitrous acid. Additional studies show that both pathways involve the intermediacy of acetylhydrazine and glutamylhydrazine as carriers of a hydrazine moiety. Furthermore, the results from experiments with ¹⁵N-labeled precursors preclude the intermediacy of free hydrazine in either of these pathways. These findings reveal a biosynthetic strategy that is completely different from the biosynthesis of some N-N bond

containing natural products where the N-N bond is formed from two nitrogen atoms already located on the scaffold of the final product. The results described here provide highly novel and important insights regarding the biosynthesis of some N-N bond containing natural products. They will be of great interest to investigators studying natural product biosynthesis. The experimental and supplementary data included with the manuscript provide compelling evidence in support of the conclusions drawn by the authors.

REPLY: We appreciate the kind words and the recognition that this work is a major step forward in understanding the biosynthetic logic of an important class of compounds.

Wilfred A. van der Donk
Investigator
Richard E. Heckert Professor of Chemistry

University of Illinois at Urbana-Champaign
Department of Chemistry
600 South Mathews Avenue, Box 38-5, Urbana, IL 61801

REVIEWERS' COMMENTS:

Reviewer #1 (Remarks to the Author):

The authors have addressed all previous concerns.

Reviewer #2 (Remarks to the Author):

I am satisfied with the revised manuscript and response to all reviewers. I recommend accept I the current form.